# A Closer Look at the Intervention Procedure of Concept Bottleneck Models

**Sungbin Shin**
POSTECH
ssbin4@postech.ac.kr

**Yohan Jo**[*]
Amazon
jyoha@amazon.com

**Sungsoo Ahn**
POSTECH
sungsoo.ahn@postech.ac.kr

**Namhoon Lee**
POSTECH
namhoonlee@postech.ac.kr

## Abstract

Concept bottleneck models (CBMs) [1] are a class of interpretable neural network models that predict the target label of a given input based on its high-level concepts. Unlike other end-to-end deep learning models, CBMs enable domain experts to intervene on the predicted concepts at test time so that more accurate and reliable target predictions can be made. While the intervenability provides a powerful avenue of control, many aspects of the intervention procedure remain underexplored. In this work, we inspect the current intervention practice for its efficiency and reliability. Specifically, we first present an array of new intervention methods to significantly improve the target prediction accuracy for a given budget of intervention expense. We also bring attention to non-trivial yet unknown issues related to reliability and fairness of the intervention and discuss how to fix these problems in practice.

## 1 Introduction

Recent advances in deep learning have made remarkable success in various areas of machine learning [2, 3]. However, standard neural network models are still not easily explainable, making it difficult for humans to understand the rationale behind or improve upon the decision-making process of these models. To tackle this issue, various sets of interpretable models have been proposed, for example, using concept activation vectors [4, 5], finding the contributions of each pixel to the image classification [6, 7], or building intrinsically interpretable architectures [8].

Concept bottleneck models (CBMs) are among these to empower interpretability [1, 9, 10, 11, 12, 13]. Unlike standard end-to-end models, CBMs seek to learn two-step models that utilize high-level concepts in making their final predictions: it first predicts human-understandable concept values for a given input using the concept predictor $g$, and then the subsequent prediction is made for the target task using the target predictor $f$; for example, classify the species of a bird image based on its wing pattern or leg color rather than straight from the input image (see Figure 1a).

Revisited recently by [1], this classic idea further facilitates human-model interaction in addition to plain interpretability, in that it allows one to *intervene* on the predicted concepts at test time, such that the subsequent target prediction can be made based on the rectified concept values. By intervening on the incorrectly predicted concepts and updating their values to correct ones, CBMs can potentially make more accurate and reliable prediction results.

---

[*]This work has no relationship to Amazon.

2022 Trustworthy and Socially Responsible Machine Learning (TSRML 2022) co-located with NeurIPS 2022.

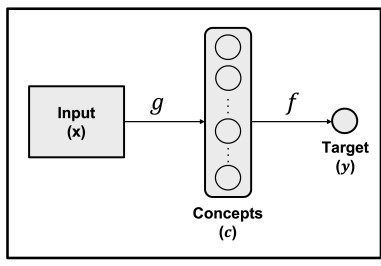

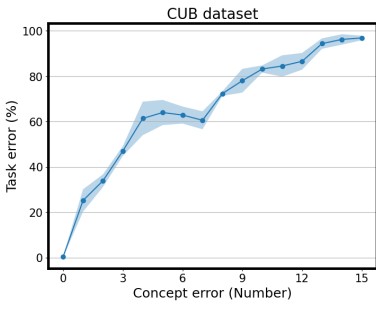

(a) standard concept bottleneck models

(b) concept vs. task errors

Figure 1: (a) Given an input data a CBM first predicts high-level concepts ($g : x \rightarrow c$), and then based on which it makes final prediction for the target task ($f : c \rightarrow y$). (b) The task error quickly increases with more errors in concept prediction. For example, making one concept prediction wrong (*i.e.*, concept error in number is at 1) leads to about $25\%$ increase in the final task error; in other words, fixing a small number of mis-predicted concepts has the potential to significantly improve the performance of CBMs.

Surprisingly, however, many aspects of the intervention procedure, for example that specifies which concept to intervene first, have not been studied much in the literature despite its potential implications in practice. Previous works have mostly focused on either increasing the final prediction accuracy of CBMs [10, 11], removing the impact of confounding information [9], or analyzing the information leakage problems [12, 13]. Indeed we find that intervening on mis-predicted concept values can have a potentially critical impact on the final task accuracy (see Figure 1b). Considering the high cost of the intervention process, *i.e.*, having domain experts go over each concept, this result further indicates the necessity of more effective and efficient intervention procedures to ensure the utility of CBMs.

In this work, we first evaluate different intervention methods and observe that the existing method, which determines the concept intervention order at random, can be improved significantly. While evaluating these methods, we find that task error can increase with more intervention, which makes the procedure less reliable. Furthermore, we observe that current pre-processing procedure adopted in some datasets can have a negative effect on model fairness. To address these issues, we suggest some possible solutions which may help to build trustworthy models.

## 2 Related work

There have been many attempts to improve the limitations of the original CBMs [9, 10, 11]. Among them, a recent work by Zarlenga *et al.* [11] proposes Concept Embedding Models (CEMs), which exploit a specific technique called *RandInt* to make the intervention more effective at test time. To be specific, this strategy randomly intervenes on some concepts at training time with the pre-defined probability. Our work differs from this one as we focus on improving the intervention method itself when pre-trained CBMs are already given and we cannot change the training procedure of the models. Other previous works try to increase the performance of the model either by removing the impact of confounding information [9] or adding unsupervised concepts [10], but they also do not directly address the intervention procedure. As of our knowledge, our work is the first to analyze CBMs in terms of intervention, one of the most important features of the models.

In addition, another line of works aims at analyzing the non-trivial problems lying in CBMs. Specifically, [12] and [13] argue that information leakage can occur, *i.e.*, the model can encode more information in the concept predictor $g$ than ground-truth concepts specified in the dataset so as to increase the performance. This problem decreases the interpretability of the model and makes it hard to understand the reason behind the model's prediction. In a similar vein, we further analyze what can potentially decrease the reliability or fairness of CBMs and discuss some possible solutions.

# 3 Methods

In this section, we introduce the set of intervention methods we consider to be applicable to CBMs. To be specific, each method defines a criterion for determining the order of concepts to be intervened on. We use $g(\cdot)$ and $f(\cdot)$ to denote the concept predictor and the target predictor, respectively. We also use $\hat{c}_i \in [0, 1]$ and $c_i^{\text{GT}} \in \{0, 1\}$ to denote the predicted and the ground-truth value of the $i$-th concept $c_i$. Note that we are assuming each concept as a binary variable representing the existence of the attribute as they are usually in most real-world datasets experimented in [1, 11, 13].

**Random**  Following the original work [1], this method randomizes the order of the intervention. To be specific, it samples each concept from a uniform distribution to determine the order.

**Concept uncertainty**  This method first intervenes on the concept with the highest predictive uncertainty from the concept predictor $g(\cdot)$. It is inspired by active learning [14], whose goal is to increase the performance of the model with the minimum amount of labeling cost. Such an idea applies to the intervention of CBMs since it also aims at decreasing task error with the minimum amount of concept labeling cost. Based on [15, 16], this method measures the uncertainty of a binary concept prediction $\hat{c}_i$ as the inverse of its distance to the value 0.5, *i.e.*, it first intervenes on the concept with the maximum value of $1/|\hat{c}_i - 0.5|$. Note that other popular uncertainty estimation methods like maximum entropy [17], least confidence [18], or smallest margin [19] all leads to the same intervention orders when the concepts take binary values.

**Concept importance**  This method prioritizes the intervention on the concept which is the most "important" for class label prediction, since mis-predicting such a concept results in a significant increase in the task error. In the case when $f$ is 1-layer MLP, we define the concept importance as the product of the predicted concept value and the sum of related weights, *i.e.*, $\hat{c}_i \sum_{j=1}^{M} |w_{ij}|$, inspired by Grad-CAM [6]. Here, $w_{ij}$ is the weight of the $i$-th concept to the $j$-th class in $f$ and $M$ is the number of target classes.

**KL divergence**  Next, we consider intervening on the concept with the most impact on the target predictive distribution when the concept value changes to 0 after the intervention. To this end, this method computes the new distribution of target class prediction after intervening on $c_i$ with 0 as follows:

$$\hat{y}_{\text{tti}}^i = \text{Softmax}(f(\hat{c}; c_i = 0)).$$

Here, $f(\hat{c}; c_i = 0)$ denotes the output of $f$ when $i$-th concept is manually set to 0. Intuitively, this represents how the target prediction will change when $i$-th concept does not exist anymore. Then, this method utilizes Kullback-Leibler (KL) divergence [20] to measure the change between the target predictive distributions before and after the intervention, *i.e.*, $D_{KL}(\hat{y}_{\text{tti}}^i \| \hat{y})$. Finally, it first intervenes on the concept with the largest amount of change.

**Smallest margin**  This method measures the uncertainty in the target class with respect to the $i$-th concept intervention with 0. Among the various candidates for uncertainty estimation, we consider the smallest margin [19]. This method prioritizes intervening on the concept $c_i$ with the smallest margin of $(y_1^*)^i - (y_2^*)^i$ where $(y_1^*)^i$ and $(y_2^*)^i$ denote the first and second largest target class probability in $\hat{y}_{\text{tti}}^i$. Thus, the method first selects the concept that makes the target prediction to be most uncertain when it disappears from the image. Note that we also considered other scores such as maximum entropy [17] and least confidence [18], but the smallest margin performed the best (see Appendix C for the results).

**Largest concept loss**  This method intervenes on the concept with the largest concept prediction loss. This way, it simulates the "oracle" metric for measuring the concept uncertainty. To be specific, it intervenes on the concept in the decreasing order of $|c_i^{\text{GT}} - \hat{c}_i|$. However, we remark that the ground-truth concept values are not available at test time, so it is hard to use this method in practice. Nonetheless, by comparing with the other methods which do not use the ground-truths, it can tell us how much a specific intervention method can be improved.

## 4 Experimental Setup

**Dataset** All the experiments are conducted using the CUB dataset [21] which is the standard dataset used to study CBMs. There are $5994$ and $5794$ examples for train and test sets in total, in which each example consists of the triplet of (image $x$, concepts $c$, label $y$) of a bird species. All the concepts have binary values; for example, the '`wing color:black`' for a given bird image can be either $1$ (for true) or $0$ (for false). Following previous works [1, 10, 11], we perform so-called majority voting as pre-processing so that images of the same class always have the same concept values; for example, if more than half of the crow images have true value for the concept '`wing color:black`' then this process converts all concept labels for images belonging to the crow class to have the same true value. Since the original concept labels are too noisy, this procedure helps to increase the overall performance. However, it can be potentially harmful to model fairness in some cases as we will cover later in Section 6. Also after removing too sparse concepts, we end up with $112$ out of $312$ concepts remaining (see Appendix D for the details).

**Model** Following the standard setup as in [1], we use Inception-v3 [22] pretrained on Imagenet [23] for the concept predictor $g$ and 1-layer MLP for the target predictor $f$ respectively, while both networks are trained with the same training hyperparameters as in [1]. We show the results of the Independent CBM ($g$, $f$ are trained separately) in the main results. The results of Sequential ($f$ is trained after finishing training $g$) or Joint models ($g$, $f$ are trained simultaneously) with or without the sigmoid activation function can be found in Appendix F (see Appendix A for model details). We report task error as the average over $5$ training runs with different random seeds.

**Group vs. Individual interventions** In previous works the intervention is done group-wise in which concepts within the same group (derived from the dataset collection process) are intervened on altogether; for example, the values for '`wing color:black`' and '`wing color:white`' are updated at once when intervened. However, we find that the concepts in the CUB dataset are not always mutually exclusive, and multiple concepts in the same group can have true values at the same time. Since intervening on the related group together requires almost the same amount of effort as intervening on the same number of concepts across different groups, we also evaluate individual-wise intervention by which concepts are intervened one at a time.

## 5 Main Results

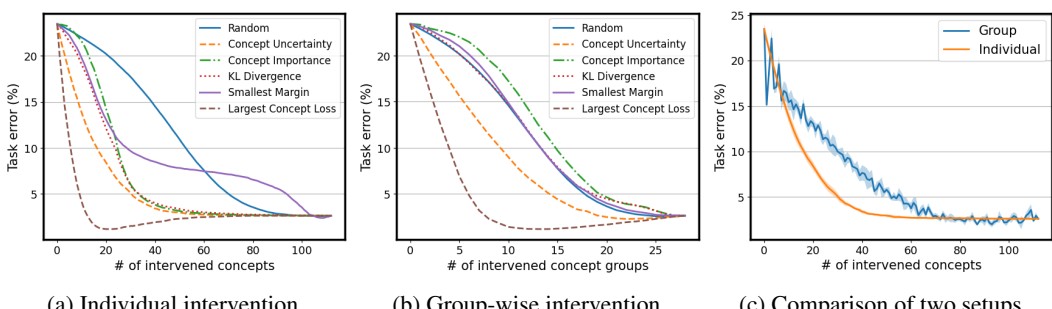

(a) Individual intervention.   (b) Group-wise intervention.   (c) Comparison of two setups.

Figure 2: (a) Comparison of different methods of individual intervention. (b) Comparison of different methods of group-wise intervention. (c) Comparison of individual and group-wise interventions (using concept uncertainty) for the effective number of intervened concepts.

Here we report evaluation results for different intervention methods developed in Section 3. For the random method, we report the average task error over 5 randomly generated intervention orders. As seen in Figure 2a, the current practice of random intervention is easily outperformed by the alternatives. Notably, utilizing concept uncertainty information helps to improve the target prediction quite significantly. Specifically, correcting $12.5\%$ ($14$ out of $112$) of concepts decreases the task error almost by half while the random method requires $42.9\%$ ($48$ out of $112$) corrections to achieve the same error rate. Other methods consistently outperform the random method as well, except for the smallest margin which becomes less effective later. Overall, we find that the intervention procedure can be much more effective than as it is with the naive current practice.

In group-wise intervention, we calculate the mean of the scores for each concept, *e.g.*, uncertainty or concept importance, within the group to determine the order of intervention. As seen in Figure 2b, concept uncertainty method is still more effective than the random method. Specifically, random intervention needs 42.9% (12 out of 28), while concept uncertainty method needs 28.6% (8 out of 28) of the groups to be intervened to decrease the task error by half. However, unlike in individual intervention, other methods do not outperform the random method. We suspect that calculating the mean of the scores in group-wise intervention loses some discriminative information, and perhaps a different surrogate needs to be designed.

In addition, we find that group-wise intervention is less effective than individual intervention with the same budget of intervention expense (see Figure 2c). To better understand this result, it is worth noting that we cannot provide rich information when only allowed to intervene concepts within the same group at a time. By choosing the same number of concepts across different groups, we can intervene on the concepts more effectively with the same effort.

Although the uncertainty method shows promising results, we also notice a significant gap from the largest concept loss method. Specifically, the latter only needs $4.5\%$ (5 out of 112) concepts to decrease the task error by half in individual intervention. Nonetheless, the largest concept loss method is unrealistic in practice as the model does not have access to the ground truth concept values to determine the order of intervention. We leave closing this gap as a future work.

## 5.1 Ablation: Counter-intuitive aspect of the intervention

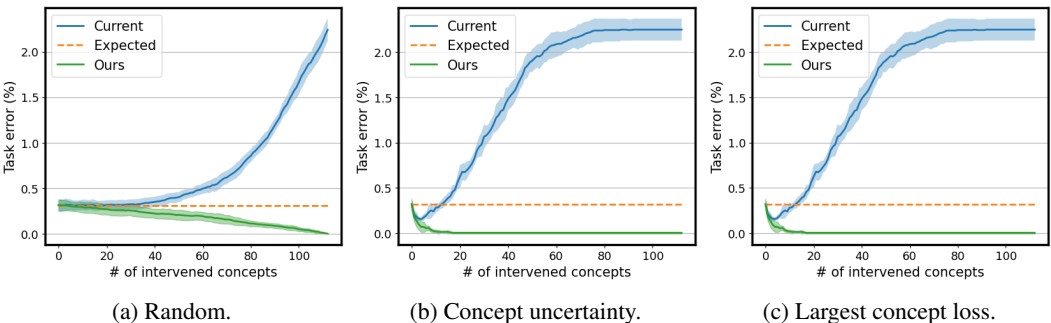

(a) Random.                  (b) Concept uncertainty.                  (c) Largest concept loss.

Figure 3: When all concepts are predicted correctly, intervention is expected to at least keep the task error constant (Expected). Contrary to the expectation, however, rectifying the concept values by intervention increases the task error when following the current protocol (Current). By not intervening on the concept values when the related parts are invisible, intervention can decrease the task error consistently (Ours). Shaded regions are the standard deviations over 5 different random seeds.

It is observed in Figure 2a that the task error does not always decrease with more intervention; *i.e.*, see the largest concept loss method for which the error starts to increase after about twenty concept intervention counts. This result is unexpected and counter-intuitive since intervention is supposed to improve the prediction performance. To better understand, we set up an ablation experiment where we have the images for which all their concepts are correctly predicted (*i.e.*, $100\%$ concept prediction accuracy); in this case, intervention is expected at least not to increase the task error ('Expected' in Figure 3). For those images, however, the task error keeps on increasing as more concepts get intervened, and when all concepts are intervened at the end, the prediction error reaches to more than seven times as much as with no intervention ('Current' in Figure 3). This reveals that intervention, as opposed to how it is believed, can actually degrade the subsequent target prediction performance, questioning its reliability.

We realize that this happens due to the specific intervention protocol used in [1] for making the intervention more realistic. Since intervening on the concepts correctly is nearly impossible when the related part is invisible from the image, the authors assume that domain experts would consider the concepts to be false or non-existent (value of $0$) in this case; for example, if the wing is invisible, concept 'wing color:black' changes to $0$ at intervention regardless of its ground-truth value. Thus, when $c_i^{\text{GT}} = \hat{c}_i = 1$, *i.e.*, the model correctly predicts the concept value as $1$, but the related part is not visible, intervention can rather increase the task error as it makes the predictions to be incorrect.

However, we find that simply not intervening on those concepts, *i.e.*, not changing the predicted values when the related part is invisible, can consistently decrease the task error when all concepts are correctly predicted ('Ours' in Figure 3). This technique also prevents the task error from increasing when using the largest concept loss method and can make the intervention more effective in general. (see Appendix E).

## 6 Fairness problem lying in majority voting

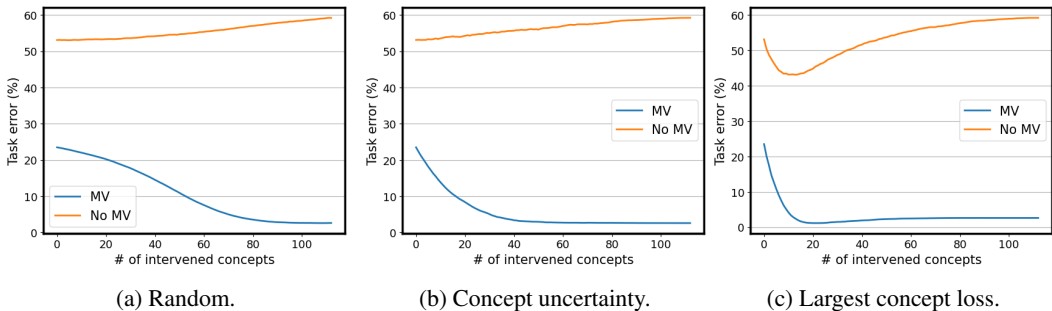

|  |  |  |
|---|---|---|
| (a) Random. | (b) Concept uncertainty. | (c) Largest concept loss. |

Figure 4: Comparison of test-time intervention results with and without using majority voting. **Left, Mid**: When we do not use majority voting, intervention does not decrease task error at all with random or concept uncertainty method. **Right**: Even with the largest concept loss method, intervention does not reduce the task error as much as when we use majority voting, and the error rather starts to increase after about 15 concepts intervened. See Appendix B for the training details.

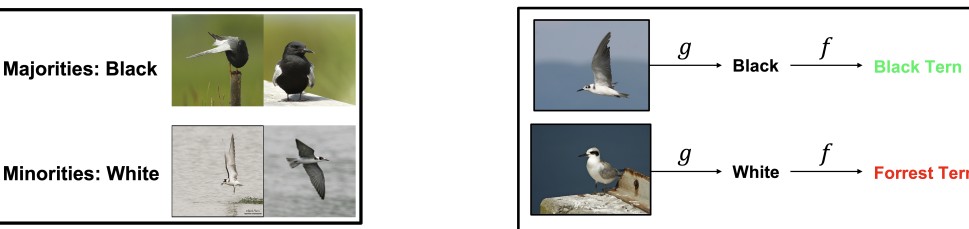

|  |  |
|---|---|
| (a) Majorities and minorities. | (b) Prediction in the minorities. |

Figure 5: An example where majority voting can be harmful to model fairness. **Left**: In 'black tern' class, a majority of the images have black underparts color, but some minorities have white underparts color. **Right**: At test time, if another minority comes in, the label prediction is correct only when the predicted concept value is the same as majority voted value.

Since the concept labels in the CUB dataset are noisy, previous works [1, 11] use majority voting to increase the task accuracy and the intervention effectiveness. When training CBMs with noisy labels without majority voting, we indeed observe that the task error is too high and intervention does not effectively reduce the error as seen in Figure 4. Despite its advantages, we find that majority voting can have a negative effect on model fairness. As a concrete example, we observe that the majority images of 'black tern' class have black underparts color, but some minorities in this class have white underparts color (Figure 5a). Majority voting changes the ground-truth underparts color of the minorities as black, which leads to the model being trained to predict black concept value from minorities. When another minority image comes in at the test time, we observe that the target class prediction is biased. In other words, the image is classified correctly only when the predicted concept is identical to the majority-voted value (black) and it is not the case when the predicted value is the original minority value (white) as seen in Figure 5b.

This phenomenon can be potentially harmful to model fairness when the dataset contains sensitive attributes. Some recent works on CBMs [11, 24] experimented on the CelebA dataset, which contains attributes related to gender ('male', 'no beard') or ethnicity ('blonde hair', 'black hair'). Since the experiments are conducted with synthetic class labels without majority voting, target class

prediction is unlikely to be biased as in the previous examples of the CUB dataset in these works. Nevertheless, applying majority voting on this kind of dataset containing sensitive attributes might lead to biased prediction and have a potentially negative effect on model fairness. Some possible approaches to prevent such scenarios include not containing sensitive attributes in the concept layer in CBMs or not applying majority voting even if they are included. We leave further investigations about the effect of majority voting on model fairness in CBMs as future research.

# 7 Conclusion

In this work, we analyze CBMs in terms of the test-time intervention, which was missing in the previous works. First, we evaluate different intervention methods which show promising results in decreasing task error with the same concept labeling cost. Furthermore, we observe that existing intervention protocol or pre-preprocessing procedure can make CBMs unreliable and unfair in some cases and discuss the way we can remove or mitigate the problems. We hope these findings can help future researchers to build more interpretable and responsible models.

# Acknowledgement

This work was partly supported by Institute of Information & communications Technology Planning & Evaluation (IITP) grant funded by the Korea government (MSIT) (No.2019-0-01906, Artificial Intelligence Graduate School Program (POSTECH) and No.2022-0-00959, (part2) Few-Shot learning of Causal Inference in Vision and Language for Decision Making) and National Research Foundation of Korea (NRF) grant funded by the Korea government (MSIT) (2022R1C1C1013366, 2022R1F1A1064569).

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

## A  Different ways to train CBMs

Let $g : \mathbb{R}^d \to \mathbb{R}^k$ and $f : \mathbb{R}^k \to \mathcal{Y}$ be the concept predictor and the target predictor respectively. The original work [1] introduces 3 different ways to train the two networks, which we describe in this section. Here, we use the notation of $N$ for the number of data points, $\sigma$ for the sigmoid activation function, and $l_c, l_y$ for the loss functions.

The first one is the independent model, where two networks $g$ and $f$ are trained separately from each other. Specifically, two networks are trained to minimize the following objectives respectively.

$$L_g = \frac{1}{N} \sum_{i=1}^{N} l_c(\sigma(g(x^{(i)})), c^{(i)}), \ L_f = \frac{1}{N} \sum_{i=1}^{N} l_y(f(c^{(i)}), y^{(i)}).$$

The second one is the sequential model, where we first train $g$ and then train $f$ after finishing training $g$. Specifically, they are trained to minimize the following objectives respectively.

$$L_g = \frac{1}{N} \sum_{i=1}^{N} l_c(\sigma(g(x^{(i)})), c^{(i)}), \ L_f = \frac{1}{N} \sum_{i=1}^{N} l_y(f(g(x^{(i)}), y^{(i)})$$

Note that $f$ now takes predicted concept value $\hat{c} = g(x)$ as input rather than the ground-truth $c$.

The final one is the joint model, where we train $g$ and $f$ simultaneously to minimize the following single objective.

$$L_{gf} = \frac{1}{N} \sum_{i=1}^{N} \{\lambda l_c(\sigma(g(x^{(i)})), c^{(i)}) + l_y(f(g(x^{(i)}), y^{(i)}))\}$$

Here, hyperparmater $\lambda$ determines the trade-off between $l_c$ and $l_y$. We can insert the sigmoid activation function in the target predictor $f$ as follows which can make intervention more effective as found in [1].

$$L_{gf} = \frac{1}{N} \sum_{i=1}^{N} \{\lambda l_c(\sigma(g(x^{(i)})), c^{(i)}) + l_y(f(\sigma(g(x^{(i)})), y^{(i)})\}$$

## B  Training details of when not using MV

For training the concept predictor $g$ and the target predictor $f$ when not using majority voting, we used the same architecture, Inceptionv3 pretrained on the Imagenet for $g$. For $f$, we used the 2-layer MLP whose hidden layer is of dimension $200$ so that it can describe more complex functions. We searched the best hyperparameters for both $g$ and $f$ over the same sets of values as when using majority voting in [1]. Specifically, we tried initial learning rates of $[0.01, 0.001]$, constant learning rate and decaying the learning rate by $0.1$ every $[10, 15, 20]$ epoch, and the weight decay of $[0.0004, 0.00004]$. After finding the optimal values of hyperparameters whose validation accuracy is the best, we trained the networks with the same values again over 5 different random seeds.

## C  Evaluation of other intervention methods based on the target class uncertainty

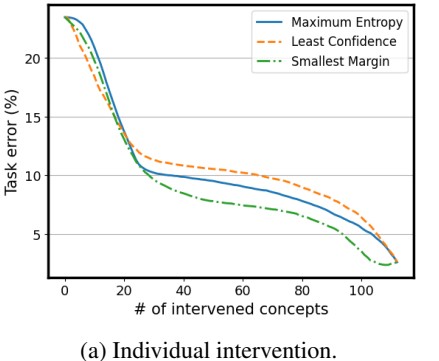

(a) Individual intervention.

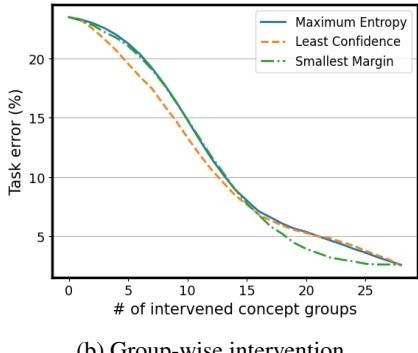

(b) Group-wise intervention.

Figure 6: Results of intervention methods measuring the uncertainty in the target class with respect to the $i$-th concept intervention with 0. All the methods work similarly, but the smallest margin is the most effective one in the individual intervention.

We define the maximum entropy and least confidence method as follows.

**Maximum entropy**   To measure the uncertainty in the target class with respect to the $i$-th concept intervention with 0, this method utilizes the maximum entropy [17]. Specifically, it first intervenes on the concept $c_i$ with the largest value of $H(\hat{y}_{\text{tti}}^i)$ where $H(X) = \mathbb{E}[-\log p(X)]$ is the function measuring the entropy in the probability distribution and $\hat{y}_{\text{tti}}^i$ is defined in Section 3.

**Least confidence**   This method utilizes the least confidence [18] to measure the uncertainty in the target class with respect to the $i$-th concept intervention with 0. Specifically, it first intervenes on the concept $c_i$ with the smallest value of $(y_1^*)^i$, the largest class probability in $\hat{y}_{\text{tti}}^i$.

As seen in Figure 6, the smallest margin method is slightly more effective than the others in the individual intervention, though the difference is not huge.

## D  Details about CBMs

**Concept Bottleneck Models**   Concept Bottleneck Models(CBMs) [1] are a class of interpretable deep neural networks which utilize high-level concepts to predict the target label. Borrowing the

notations from [1, 12], let $i$-th data point be $(x^{(i)}, c^{(i)}, y^{(i)})$ where $x, c$, and $y$ each correspond to the input image, binary concepts, and the target label. Here, $x \in \mathbb{R}^d, c \in \{0, 1\}^k, y \in \mathcal{Y}$ where $d$ is the dimension of the input images, $k$ is the number of concepts, and $\mathcal{Y}$ is the target space. Though predicting $c$ and $y$ can either be a classification or regression task, we only focused on the case when $c$ and $y$ are classification tasks in this paper following the dataset we used for the experiments. The model consists of the two networks $g : \mathbb{R}^d \to \mathbb{R}^k$, the concept predictor which predicts concepts from the input image, and $f : \mathbb{R}^k \to \mathcal{Y}$, the target predictor which predicts the target label from the concepts. At test time, we predict the target label using the two networks as $\hat{y} = f(g(x))$.

**Test-time Intervention** Unlike other interpretable models, CBMs support test-time intervention so that domain experts can change the concept values at test time, which may help the model make correct target label predictions. The effectiveness of the intervention can be measured by how fast the task error decreases, *i.e.*, the decrease in the task error with the same number of intervened concepts, where task error indicates the proportion of the misclassified images.

**Pre-processing** To build CBMs with the CUB dataset, we follow the pre-processing of [1] and perform majority voting since the original concept labels are too noisy (see Section 4 for details). After this process, we remove too sparse concepts whose value is present in less than 10 classes. The authors of [1] explain that including these sparse concepts in the concept layer makes it hard to predict their values as the positive training examples are too scarce. Since high concept error can further lead to an increase in task error, we do not use these concepts when training the model following their setup.

# E    Results of ablation study

Our simple technique introduced in Section 5.1 not only increases the reliability of the intervention, but can also make it more effective. As seen in Figure 7, task error decreases more with the same number of intervened concepts consistently in the 3 intervention methods. Note that in the largest concept loss method, task error now keeps on decreasing and does not increase after 20 intervened concepts.

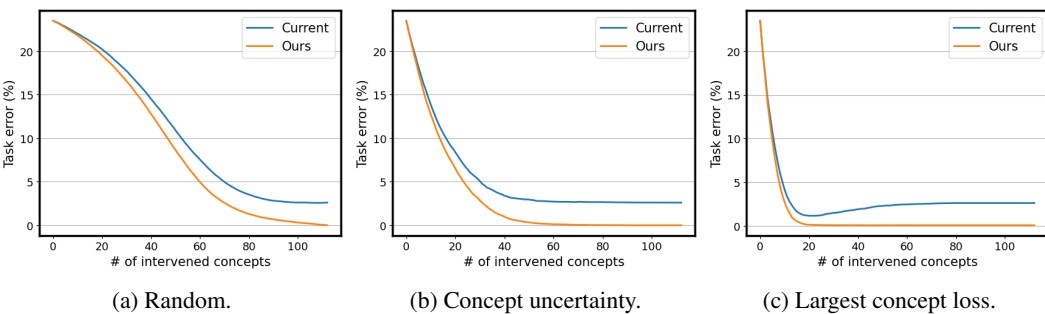

(a) Random.              (b) Concept uncertainty.              (c) Largest concept loss.

Figure 7: By not intervening on the concepts whose related parts are not visible from the image, intervention is more effective consistently in all the methods.

# F    Results of sequential and joint models

Here, we present the results of various intervention methods in sequential and joint CBMs. We also experimented on the joint models with the sigmoid activation function, as it is found in [1] that this improved the intervention effectiveness of CBMs. For sequential (Figure 8) and joint CBMs without sigmoid activation function (Figure 9), lots of intervention methods are similar to the random method. We suspect that this is because intervention is less ineffective in these models considering that concept predictor $g$ is trained with the predicted concept values $\hat{c}$ rather than the ground-truth values $c$. Nevertheless, we find that the largest concept loss and concept uncertainty method consistently show promising results. For the joint models with sigmoid activation function (Figure 10), lots of methods are more effective than random method in individual intervention as

similar to the independent models. We observe that inserting the sigmoid function indeed helps the intervention procedure.

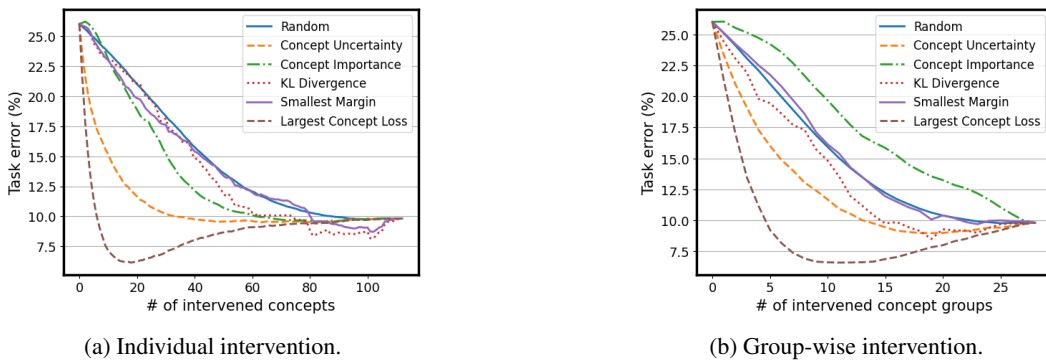

(a) Individual intervention.

(b) Group-wise intervention.

Figure 8: Results of the various intervention methods in sequential CBMs.

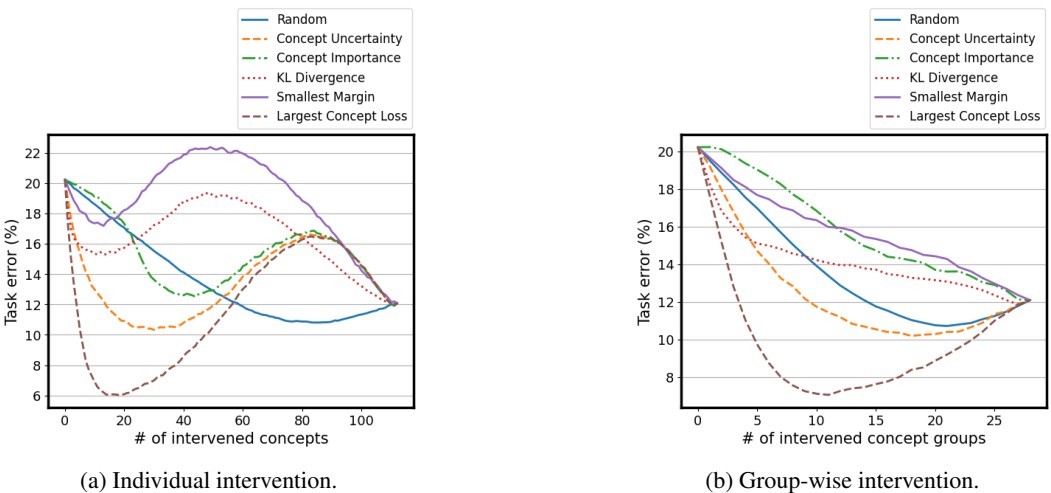

(a) Individual intervention.

(b) Group-wise intervention.

Figure 9: Results of the various intervention methods in joint CBMs without sigmoid activation function.

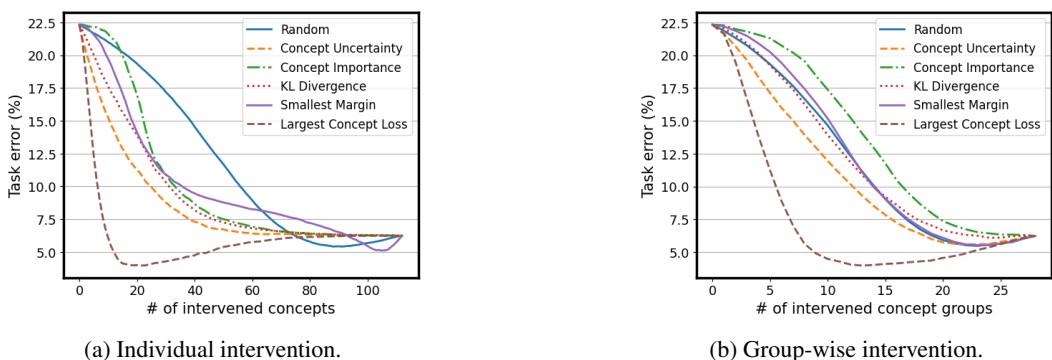

(a) Individual intervention.

(b) Group-wise intervention.

Figure 10: Results of the various intervention methods in joint CBMs with sigmoid activation function.

