# OpenReview forum: "A Closer Look at the Intervention Procedure of Concept Bottleneck Models"
_NeurIPS.cc/2022/Workshop/TSRML — TSRML2022_

### Official Review · Reviewer_g799 · 2022-10-13
**A Comparison of Different Intervention Procedures for Concept Bottlenecks**

**Overall Rating:** 7

**Summary:**

This paper considers several intervention procedures for concept bottleneck models. The experiments on the standard CUB benchmark show that more efficient strategies exist than naively intervening on a subset of the concepts chosen uniformly at random.

**Strengths:**

* To the best of my knowledge, the intervenability of CBMs has not been studied in detail in previous works, mainly focusing on the problems of leakage and confoundedness.

* The paper is well-written; the results are presented visually and are easy to follow.

* The proposed intervention strategies are sensible and likely helpful when implementing CBMs in practical settings.

**Weaknesses:**

* One limitation of the experimental setup is the lack of results for sequentially and jointly optimised CBMs. Since these models were observed to be less intervenable in the original paper, I wonder if the differences between different intervention strategies are still present.

* The use of the CUB dataset as the benchmark might be problematic. Class-level concepts were used in the original CBM paper (obtained by majority voting). This work implements the same preprocessing to avoid concept noisiness. However, this step can be an impractical simplification for specific applications, e.g. in healthcare. It would be good to incorporate another dataset with instance-level concepts, e.g. by generating a synthetic dataset.

* For the reasons described above, I believe that the fairness issue described in Section 6 is an artefact of inappropriate preprocessing rather than of the CBM itself. Hence, the problem is somewhat exaggerated.

**Overall Recommendation:**

The contribution of this paper is moderate: an empirical study of different intervention strategies for CBM models. This investigation and findings would be helpful for practitioners applying CBMs since some of the proposed procedures are more efficient than naive interventions on random concept subsets.

**Review Confidence:**

4: The reviewer is confident but not absolutely certain that the evaluation is correct

---

### Official Review · Reviewer_BZh3 · 2022-10-21
**Good study on intervention procedure**

**Overall Rating:** 7

**Summary:**

This paper studies the intervention practice for concept bottleneck models. Specifically, the paper discusses a range of intervention criteria and demonstrates the gap between an ideal performance and the current methods. The authors also point out the reliability and fairness issues with the intervention procedures.

**Strengths:**

This paper serves as a good initial study on the intervention strategies in CBMs. It gives an array of new intervention models and conducts experiments for both individual and group interventions to show their effectiveness. The motivations are self-evident and the proposed methods are intuitive.

**Weaknesses:**

It would be great if the authors can show some statistics of how the current CBMs contain sensitive attributes, as discussed in section 6.

**Overall Recommendation:**

I believe this paper can inspire more intervention studies for CBMs in the future.

**Review Confidence:**

3: The reviewer is fairly confident that the evaluation is correct

---

### Decision · Program_Chairs · 2022-10-23

**Decision:**

Accept

**Comment:**

Following the unanimous recommendations from reviewers, the submission is accepted.